 SCIENTIFIC CORRESPONDENCE

# Response to comment on 'Parasite defensive limb movements enhance acoustic signal attraction in male little torrent frogs'

**Longhui Zhao[1,2], Wouter Halfwerk[3]\*, Jianguo Cui[1]\***

[1]CAS Key Laboratory of Mountain Ecological Restoration and Bioresource Utilization & Ecological Restoration and Biodiversity Conservation Key Laboratory of Sichuan Province, Chengdu Institute of Biology, Chinese Academy of Sciences, Chengdu, China; [2]Ministry of Education Key Laboratory for Ecology of Tropical Islands, Key Laboratory of Tropical Animal and Plant Ecology of Hainan Province, College of Life Sciences, Hainan Normal University, Haikou, China; [3]Department of Ecological Sciences, Vrije Universiteit Amsterdam, Amsterdam, Netherlands

**Abstract** Recently we showed that limb movements associated with anti-parasite defenses can enhance acoustic signal attraction in male little torrent frogs (*Amolops torrentis*), which suggests a potential pathway for physical movements to become co-opted into mating displays (Zhao et al., 2022). Anderson et al. argue for alternative explanations of our results and provide a reanalysis of part of our data (Anderson et al., 2023). We acknowledge some of the points raised and provide an additional analysis in support of our hypothesis.

**\*For correspondence:**
w.h.halfwerk@vu.nl (WH);
cuijg@cib.ac.cn (JC)

**Competing interest:** The authors declare that no competing interests exist.

## Introduction

Physical movements have been viewed as the raw material of visual signals for many years (*Harper, 1991*; *Johnsgard, 1962*). However, little is known about how physical displays evolve and become a part of communicative systems. According to the sensory exploitation hypothesis, physical movements may be incorporated into multimodal systems if they enhance the attractiveness of male individuals (*Ryan, 1998*). Recently we showed that movements associated with anti-parasite behavior can act on female perception and may thereby affect the evolution of a multimodal display in little torrent frogs (*Amolops torrentis*) (*Zhao et al., 2022*). Anderson et al. provide some alternative interpretations for our findings, some of which we found useful for discussion and future experiments (*Anderson et al., 2023*). Their most important argument concerns our finding that females showed a strong preference for limb movements that were not strongly associated with the presence of eavesdropping parasites. Below, we respond and provide an additional analysis based on some of the suggestions made by Anderson et al.

## Results and discussion

### Underestimations of parasite-associated limb movements

Anderson et al. suggest that the two movements most strongly preferred in playback experiments by females – hind foot lifting (HFL) and arm waving (AW) – are not associated with the presence of eavesdropping parasites, whereas two other movements – wiping (W) and limb shaking (LSA) – do increase as parasite numbers increase, but are less preferred (or not preferred) by females. They base

**Table 1.** Outcomes of a generalized linear mixed model (GLMM) to investigate the effect of parasite presence on four limb movements in actively displaying males.

The overall movement and noise data used in this analysis are available in *Supplementary file 1*.

| Response | Predictor | z | P | Parameter estimates |
|---|---|---|---|---|
| | Intercept | 0.858 | 0.391 | 1.251±1.457 |
| Arm waving | Noise intensity | 0.142 | 0.887 | 0.003±0.022 |
| | **Parasite** | **3.840** | **<0.001** | **0.053±0.014** |
| | Intercept | 0.442 | 0.658 | 0.745±1.683 |
| Hind foot lifting | Noise intensity | 1.057 | 0.291 | 0.026±0.024 |
| | **Parasite** | **1.788** | **0.074** | **0.026±0.015** |
| | Intercept | 1.445 | 0.148 | 2.000±1.384 |
| Limb shaking | Noise intensity | 0.186 | 0.853 | 0.004±0.020 |
| | **Parasite** | **2.277** | **0.023** | **0.034±0.015** |
| | Intercept | 0.145 | 0.885 | 0.220±1.517 |
| Wiping | Noise intensity | 0.231 | 0.817 | 0.005±0.023 |
| | **Parasite** | **2.792** | **0.005** | **0.035±0.013** |

this suggestion on a reanalysis of our original data, but we argue that their approach is problematic for two reasons.

First, their reanalysis used samples that differed from those used in our original paper. Most importantly, they included observations from non-calling males (which had been recorded outside of a breeding context for our control videos), whereas we analysed data from actively displaying males only. Many silent males were observed to be undisturbed by parasites and produced few limb movements (Supplementary file 1 in *Zhao et al., 2022*). Thus, the inclusion of data from silent males greatly underestimates any effect. Similar to previous results, our new analysis found that all movements were produced around parasites in calling individuals (*Table 1*).

Second, the proportion of parasite-induced movements is likely to be underestimated in our original study. For parasite-induced displays, we only included movements that occurred when parasites landed on the body of frogs or moved very close to them. Although the exact distances could not be measured from our videos, given the difficult field conditions, we feel that this is an appropriate approach.

## The association between limb displays and parasites

Anderson et al. also argue that the association between limb displays and parasites could be confounded by other environmental variables, such as local climatic conditions, densities of calling males or ambient noise levels. We agree on this issue and have performed a new analysis of our observational data that includes background noise level as a covariate (see *Table 1*).

When taking additional environmental variables into account, we now find that all four movements tested on female preferences increased with parasite presence. It is important to note, however, that it is not clear how effective the limb movements, in particular a movement such as hind-foot lifting, are as an anti-predator strategy. As suggested by Anderson et al., it would be interesting to control the amount and type of parasites around calling frogs experimentally in order to obtain a better understanding of limb movements as an anti-parasite function. Mechanistically, different limb movements may be produced (somewhat) simultaneously (for example, when they are controlled by the same neuronal and/or physiological pathway). Theoretically, some movements may have evolved after females started to pay attention, and thus immediately functioned in mate attraction. If such movements recruited (part of) the same mechanism, then we would also expect to find a relation with parasite pressure, even if the movement lost or never possessed an anti-parasite function.

## The role of male–male interactions is another interesting topic for future studies

The role of limb displays in male–male competition can be easily observed and recorded in the wild, whereas observing or manipulating male–female interactions is challenging in many species. Anderson et al. summarize studies on limb displays in torrent frogs and show that data on female choice behavior is rare, but we disagree with the statement that this means that female choice behavior has a smaller role than male–male competition. Only very few studies have addressed multimodal displays in the context of intra- and intersexual interactions simultaneously, but when they do, they typically show that males and females pay attention to similar cues (*Halfwerk et al., 2014*; *James et al., 2022*). In animal communication, it is generally assumed (although perhaps wrongly) that male responses to playback reflect female responses and thus can be used to study signal evolution in a broad comparative framework. It would be interesting to test whether male torrent frogs respond similarly to limb movements that are associated with parasite pressure, as this would provide a much easier way to assess how these movements and responses evolved across the phylogeny.

## Conclusions

Our data on males is observational and does not provide evidence for causal relationships, a point on which we agree with Anderson et al. The novelty of our paper lies in the fact that we propose a novel hypothesis of how and why multimodal displays evolve, namely via a process of co-option and possibly sensory exploitation. Studying the evolutionary history of multimodal displays, such as the limb movements produced during calling in torrent frogs, requires a phylogenetic comparative approach. For such a study, data on male–male interactions might be more useful than data on male–female interactions, given the challenges of collecting such data in the field. Finally, our data on female choice is highly novel because we show that several limb movements enhance the attractiveness of a calling male, which is a prerequisite for the co-option of cues into multimodal displays.

## Materials and methods

Anderson et al. pointed out that different males had great variation in the number of limb movements, which may bias our original statistics. They tested whether the probability of limb movements increased to a significant extent in response to parasite presence. However, they included silent males outside of a breeding context in their reanalysis, and may therefore underestimate the effect size. This study focused on breeding behaviors in which only calling males are relevant. Moreover, parasite presence was mediated by calling behaviors because parasites often used acoustic signals to find frogs. Thus, only calling individuals were included in our analysis.

We performed a new analysis using generalized linear mixed models (GLMMs) in R (v.4.2.0) to test the effect of parasites on limb display. Models using a Poisson distribution and log-link function were constructed using the package *lme4*. In each model, the frequency of one of four types of limb display was added as the dependent variable, while parasite number was included as fixed factor and overall movements as random factor. The overall movements represented the total number of all types of limb display. Anderson et al. proposed that individuals with the same proportion of a specific movement may have different impact if the absolute total number of limb movements is different. So the overall movements were treated as random factor to avoid potential statistical bias. Furthermore, to control for environmental variation, noise level at the male display site was added as covariate. The noise intensity of the habitat may vary in different locations and could affect the production of limb movements, as pointed out by Anderson et al. After finishing video recording, we used a sound level meter (AWA 5661, Hangzhou Aihua Instruments Co., Hangzhou, China) to measure the background noise level for each individual at the position of its head, and used these data in our new analysis. A likelihood ratio test ($\chi^2$=0, df = 1, $P$=1) did not show a significant reduction in explanatory power when we compared a full model (which included all terms of interest) with a collapsed model (which excluded a random factor of male ID). Only overall movement was treated as the random factor in order to avoid singularities (*Table 1*). Individual and environmental variation may affect parasite distribution and limb display. The GLMM evaluated the relationship between the frequency of parasite visit

and the number of limb movements, while controlling for individual and environmental variation, and was therefore appropriate to test whether the presence of parasites increased limb movement.

---

# Additional information

### Funding

| Funder | Grant reference number | Author |
|---|---|---|
| National Natural Science Foundation of China | 32101240 | Longhui Zhao |

The funders had no role in study design, data collection and interpretation, or the decision to submit the work for publication.

### Author contributions

Longhui Zhao, Conceptualization, Data curation, Formal analysis, Funding acquisition, Visualization, Methodology, Writing – original draft, Writing – review and editing; Wouter Halfwerk, Jianguo Cui, Conceptualization, Data curation, Visualization, Methodology, Writing – original draft, Writing – review and editing

### Author ORCIDs

Longhui Zhao ⬤ http://orcid.org/0000-0002-8675-8175
Wouter Halfwerk ⬤ https://orcid.org/0000-0002-4111-0930
Jianguo Cui ⬤ https://orcid.org/0000-0001-8746-2803

### Decision letter and Author response

Decision letter https://doi.org/10.7554/eLife.90404.sa1
Author response https://doi.org/10.7554/eLife.90404.sa2

---

# Additional files

### Supplementary files

• Supplementary file 1. The overall movement and noise data used in the new analysis reported in this paper.

• MDAR checklist

### Data availability

The overall movement and noise data used in the new analysis reported in this article are available in *Supplementary file 1*. The other data used in this analysis are available in *Zhao et al., 2022*.

The following previously published dataset was used:

| Author(s) | Year | Dataset title | Dataset URL | Database and Identifier |
|---|---|---|---|---|
| Zhao L, Wang J, Zhang H, Wang T, Yang Y, Tang Y, Halfwerk W, Cui J | 2022 | The data of parasite-induced and spontaneous displays in each limb movement for calling males, silent males and males that have females nearby | https://doi.org/10.5061/dryad.f1vhhmgzg | Dryad Digital Repository, 10.5061/dryad.f1vhhmgzg |

---

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
