## [Decision Letter]

*In the interests of transparency, eLife includes the editorial decision letter and accompanying author responses. A lightly edited version of the letter sent to the authors after peer review is shown, indicating the most substantive concerns; minor comments are not usually included.*

Thank you for submitting your article "Response to comment on 'Parasite defensive limb movements enhance acoustic signal attraction in male little torrent frogs' " to eLife for consideration as Scientific Correspondence.

Your article, and the original comment by Fuxjager and colleagues, have been reviewed by two peer reviewers (who have opted to remain anonymous), and the evaluation was overseen by Ammie Kalan as the Reviewing Editor and Christian Rutz as the Senior Editor.

We have decided to accept for publication both the original comment and your response (although your response will require some revisions; please see below). The substantive comments in this decision letter will also be published as part of your article (and likewise for the comment from Fuxjager).

The reviewers supported publication of your response to the comment article, subject to the following points being addressed in a revised version. In particular, please respond to the points raised by Reviewer #1.

*Essential Revisions:*

1) Please respond to the concerns regarding your mixed model analysis. Although use of mixed models is not a problem per se, there needs to be some careful explanation of the variables and structure of the model and how they address the particular criticisms raised in the comment (e.g., the sampling-effort issue described by Reviewer #1 below).

2) Please make sure the revised data you analyzed for your response are included in your revised submission.*Reviewer #1:*

This paper responds to some criticisms of the authors' original article about limb movements and their origin and function in a frog species. The commenters had made an alternative analysis of the data, and here the authors provide yet a different analysis of the data that comes to a different conclusion. I think the authors here do a good job of discussing the main criticisms and I appreciate their efforts to take them seriously and reconsider their own data. I think there are a few places where the rebuttals aren't particularly strong, or are still a bit unclear, which I detail below. The main one is this reanalysis of the data, which needs more explanation and justification.

First paragraph in the “Underestimations of parasite-associated limb movements” subsection: This could be an important point, but it would help to explain why it would be wrong to include recordings from non-calling males. If the question is about whether limb movements are a response to parasite presence, why does it matter whether the male is calling or not? I don't see a good justification for excluding those individuals, but if the authors have one, it would strengthen their argument. (see also first paragraph of “Materials and Methods”)

First paragraph of “Results and Discussion”: It would be helpful to reference Table 1 and the new analyses already here. When I first read this it raised the question of what would happen if a reanalysis was actually done. Later it becomes clear that there was indeed a reanalysis. But it would have made it easier to follow this argument if the results were referred to right away.

Second paragraph in the “Materials and Methods”: I didn't follow what "overall movements of different males" means, or why that would be a random effect. In general, this description of the mixed model is unclear, and this is a pretty important point since it's meant to be refuting the argument put forth in the comment. So this really needs to be clarified. I don't see why a mixed model makes sense here. It actually is pretty difficult to see how to test the hypothesis that limb displays are produced more when parasites are present than when they're absent because the null is unclear (as pointed out by the authors of the comment, who I think make a reasonable attempt to do some stats with an appropriate null). Really what we'd need is to weigh this somehow by observation effort. How many limb displays per minute are produced in the presence of parasites, and how many limb displays per minute are produced in the absence of parasites? If you don't standardize it to some kind of measure of sampling effort then it's really difficult to compare what is happening with and without parasites (it's almost an apples and oranges situation). So to summarize, it would be helpful to be very explicit about what these variables are that are included in the mixed model, to justify the use of a mixed model as opposed to some other kind of statistical test (specifically, justifying the random term), and to explain why this model is appropriate to answer the question of whether limb movements truly are more common in the presence of parasites.*Reviewer #2:*

- Does the Response respond to the criticisms made in the Comment in a way that is convincing enough to merit publication?

Yes, the responses to the Comment are helpful and somewhat convincing. But they certainly do not show that the Comments are unfounded. However, the reanalysis of their data that shows all four limb movements of the frogs are associated with parasites does tend to reject one of the comments.

- If yes, are any revisions required before the Response can be accepted for publication.

No

[Editors' note: further revisions were requested as described below.]

Clarify in the text if the model results shown in Table 1 include BOTH random effects of Male ID and Overall Movements. If not, please provide a statistic, such as Likelihood Ratio Test, to show that the models with Male ID were not significantly different from the ones with Overall Movements.

---

## [Author Response]

Essential Revisions:1) Please respond to the concerns regarding your mixed model analysis. Although use of mixed models is not a problem per se, there needs to be some careful explanation of the variables and structure of the model and how they address the particular criticisms raised in the comment (e.g., the sampling-effort issue described by Reviewer #1 below).

Thank you for your instructions. We have responded to these concerns and added more careful explanation and justification according to below comments. Please see the point-point responses below.

2) Please make sure the revised data you analyzed for your response are included in your revised submission.

Confirmed.

Reviewer #1:This paper responds to some criticisms of the authors' original article about limb movements and their origin and function in a frog species. The commenters had made an alternative analysis of the data, and here the authors provide yet a different analysis of the data that comes to a different conclusion. I think the authors here do a good job of discussing the main criticisms and I appreciate their efforts to take them seriously and reconsider their own data. I think there are a few places where the rebuttals aren't particularly strong, or are still a bit unclear, which I detail below. The main one is this reanalysis of the data, which needs more explanation and justification.

Thank you for your insightful comments. We have clarified them and replenished explanation and justification according to your detailed comments below. Please see the point-point responses below.

First paragraph in the “Underestimations of parasite-associated limb movements” subsection: This could be an important point, but it would help to explain why it would be wrong to include recordings from non-calling males. If the question is about whether limb movements are a response to parasite presence, why does it matter whether the male is calling or not? I don't see a good justification for excluding those individuals, but if the authors have one, it would strengthen their argument. (see also first paragraph of “Materials and Methods”)

This study focused on breeding behaviors in which only calling males were relevant. Moreover, parasite presence was mediated by calling behaviors because parasites often used acoustic signals to find frogs. Thus, only calling individuals were included here. This has been clarified in “Materials and Methods”. Please see revised p. 6 lines 117-120.

First paragraph of “Results and Discussion: It would be helpful to reference Table 1 and the new analyses already here. When I first read this it raised the question of what would happen if a reanalysis was actually done. Later it becomes clear that there was indeed a reanalysis. But it would have made it easier to follow this argument if the results were referred to right away.

Thank you very much. We have included the new analyses and referred to Table 1 in text.

Second paragraph in the “Materials and Methods”: I didn't follow what "overall movements of different males" means, or why that would be a random effect. In general, this description of the mixed model is unclear, and this is a pretty important point since it's meant to be refuting the argument put forth in the comment. So this really needs to be clarified.

The overall movements represented the total number of all types of limb displays. Anderson et al. proposed that individuals with a same proportion of specific movement may have different impact if the absolute total number is quite different. So the overall movements were treated as random factor to avoid potential statistical bias. This has been clarified.

I don't see why a mixed model makes sense here. It actually is pretty difficult to see how to test the hypothesis that limb displays are produced more when parasites are present than when they're absent because the null is unclear (as pointed out by the authors of the comment, who I think make a reasonable attempt to do some stats with an appropriate null).

Individual and environmental variation may affect parasite distribution and limb display. The GLMM evaluated the relationship between the frequency of parasite visit and the number of limb movement, while controlled for these factors, and therefore was appropriate to test whether the presence of parasite increased limb movement. We have explained this in text.

Really what we'd need is to weigh this somehow by observation effort. How many limb displays per minute are produced in the presence of parasites, and how many limb displays per minute are produced in the absence of parasites? If you don't standardize it to some kind of measure of sampling effort then it's really difficult to compare what is happening with and without parasites (it's almost an apples and oranges situation). So to summarize, it would be helpful to be very explicit about what these variables are that are included in the mixed model, to justify the use of a mixed model as opposed to some other kind of statistical test (specifically, justifying the random term), and to explain why this model is appropriate to answer the question of whether limb movements truly are more common in the presence of parasites.

Thank you for your good advices. In fact, we had already made such observation efforts. We observed all individuals for same duration (i.e. 10 min) and quantified which was in the absence of parasites and which was in presence of parasites. If parasites were found, we also counted the number of parasite appearance. We controlled for different factors and used GLMMs to test the relationship between parasite and limb movement. This is more comprehensive than simple classification of “absence” or “presence” category.

[Editors' note: further revisions were requested as described below.]

Clarify in the text if the model results shown in Table 1 include BOTH random effects of Male ID and Overall Movements. If not, please provide a statistic, such as Likelihood Ratio Test, to show that the models with Male ID were not significantly different from the ones with Overall Movements.

Only overall movements were included in the model. We also added a statistic, Likelihood Ratio Test, to show that the removal of ID did not show a significant reduction in explanatory power. The following text has been added:

“A likelihood ratio test (χ^2^ = 0, *df* = 1, *P* = 1) did not show a significant reduction in explanatory power when compared a full model (included all terms of interest) with a collapsing model (excluded a random factor of male ID). Only overall movement was treated as the random factor in order to avoid singularities (Table 1).”